# Generation of a Porous Scaffold with a Starting Composition in the CaO–SiO₂–MgO–P₂O₅ System in a Simulated Physiological Environment

**Lorena Grima [1], María Díaz-Pérez [1], Javier Gil [1], Daniel Sola [2,3] and José Ignacio Peña [1,*]**

[1] Instituto de Ciencia de Materiales de Aragón, Dpto. Ciencia y Tecnología de Materiales y Fluidos, Universidad de Zaragoza-CSIC, 50.018 Zaragoza, Spain; lgrima@unizar.es (L.G.); mariadiaz@unizar.es (M.D.-P.); jagilcor92@hotmail.com (J.G.)

[2] Laboratorio de Óptica, Centro de Investigación en Óptica y Nanofísica, Campus Espinardo, Universidad de Murcia, 30.100 Murcia, Spain; daniel.sola@um.es

[3] Institut für Fertigungstechnik, Technische Universität Dresden, 01062 Dresden, Germany

[*] Correspondence: jipena@unizar.es



**Featured Application: Porous scaffolds for bone-tissue growth.**

**Abstract:** Magnesium-based ceramics are involved in orthopedic applications such as bone scaffolds or implant coatings. They provide structural support to cells for bone ingrowth, but highly porous matrices cannot resist severe mechanical stress during implantation. In this study, the laser floating zone (LFZ) technique is used to prepare a dense crystalline material with composition in the CaO–SiO₂–MgO–P₂O₅ system. This material, under physiological conditions, is able to generate a porous scaffold controlled by the dissolution of the MgO phase, meeting the mechanical advantages of a dense material and the biological features of a porous scaffold. FESEM (Field emission scanning electron microscopy), XRD (X-ray Diffraction), EDS (Energy Dispersive X-rays spectroscopy), and ICP ((Inductively Coupled Plasma) analysis were carried out in order to characterize the samples before and after immersion in simulated body fluid (SBF).

**Keywords:** bioceramic scaffolds; bone regeneration; laser floating zone; magnesium oxide

## 1. Introduction

The increase in the age of the population carries greater incidence of musculoskeletal pathologies such as fractures, osteoporosis, and bone infection and tumors. The most commonly used therapies consist of the use of autografts, allografts, and xenografts, which present limitations such as the limited sources of bone, graft rejection problems, and the transmission of diseases. Implants and biomedical devices have been used to replace bones and joints, but these solutions are subject to many limitations such as fatigue, fractures, toxicity, and wear. In addition, the behavior of these implants and devices do not completely meet the requirements to which bone tissue is subjected [1–3].

In many situations, the success depends on the development of porous matrices (scaffolds) that provide the structural and mechanical support to cells for their attachment and proliferation. The material chosen as a matrix must meet be non-toxic, biocompatible, osteoproductive, osteconductive, and bioabsorbable and have sufficient mechanical properties to provide structural support during bone growth and remodeling [4–6].

Surface active silicate-based bioceramics are a subject of research on candidates for hard tissue regeneration because of their bioactivity. It is well known that wollastonite ($CaSiO_3$), larnite ($Ca_2SiO_4$), diopside ($CaMgSiO_6$), akermanite ($Ca_2MgSi_2O_7$), bredigite ($Ca_7MgSi_4O_{16}$), forsterite ($Mg_2SiO_4$),

enstatite ($MgSiO_3$), and their combinations are suitable for tissue engineering that favors implant attachment to bone tissue, as they have the ability to generate a hidroxyapatite (HAp) layer in contact with simulated body fluid (SBF) and to stimulate the proliferation and adhesion of osteoblast cells. Among the multiphasic ceramics, $Ca_3(PO_4)_2$–$CaSiO_3$ [7–9], $Ca_3(PO_4)_2$–$CaMg(SiO_3)_2$ [10], $CaSiO_3$–$CaMg(SiO_3)_2$ [11], and $Ca_3(PO_4)_2$–$CaSiO_3$–$CaMg(SiO_3)_2$ [12] eutectics, also suitable for tissue engineering, have been widely studied for their bioactivity. The SBF solution mimics human blood plasma in terms of pH and ionic concentration. The choice of one or another ceramic material depends on the kinetics of HAp deposition, the degradation rate, or the enhanced mechanical behavior if load bearing properties are required. It is also well known that the presence of Ca, Mg, and Si ions can influence proliferation and osteogenesis-related gene expressions in the different stages of osteoblastic differentiation in a way that is favorable in the process of bone remodeling [13,14].

The main disadvantages of ceramics, glasses, bioglasses, and bioactive crystals are their fragility when they have a porous structure, and their surfaces' limited bioactivity when they are completely dense [15,16]. For this reason, some researchers have designed multiphasic ceramics with the ability of generating a porous structure by the dissolution of one of the phases in the presence of SBF. Moreover, an adequate selection of the phases enables one to adjust the mechanical properties, bioactivity, ion release, and resorption rate to the specific needs of the bone. However, the size of the porous structure formed is limited to the size of the resorbable phase, which is micrometric in the case of bioeutectics [8]. Nevertheless, surface microporosity might improve the bioactivity of the scaffolds, thereby enabling the adsorption of proteins and cells to a larger surface area.

Phase dissolution occurs via the breaking of bonds catalyzed by the absorption of protons or hydroxyl ions to such bonds. In the case of Wollastonite dissolution, the release of $Ca^{2+}$ ions is accompanied by $H^+$ penetration deep into the structure, promoting silica tetrahedral condensation leading to the formation of a thin amorphous silica-rich layer at the surface. Such a layer slows down Wollastonite dissolution by decreasing the surface area exposed to the reactive fluid. This process influences the depth of the porous layer when a resorbable phase is present in a multiphasic bioceramic. Due to the consumption of protons ($H^+$), the pH increase and the newly formed negative silica layer attracts positive ions, and this gives rise to the re-adsorption of Ca, Mg, and other ions such as $HPO_4^{2-}$ and $OH^-$ from the media.

The role of $Mg^{2+}$ ions in bone remodeling, skeletal development, human metabolism, and cellular processes such as bone cell adhesion and osteoblast proliferation is well established [17,18]. Oelkers et al. [19] reviewed the olivine dissolution rates in aqueous fluids and reported an initial non-stoichiometric release of Mg and $SiO_2$ due to the equilibration of the olivine (Forsterite) surface with the liquid, and this release was followed by an increase in Mg and $SiO_2$ concentrations in the liquid linearly with time. These ions are preferably released at low to neutral pH. The dissolution rates of the Forsterite, like other Mg-silicates, appear to decrease monotonically with increasing pH. In acidic conditions, the formation of a passivating amorphous $SiO_2$ layer is more active and is favored by the high solubility of Mg, which creates an increasing Si-rich surface.

The advantage of using MgO as a soluble phase is that the breaking of Mg–O bonds is the only step in the steady-state dissolution mechanism favoring the formation of interconnected porous pathways. This fact contrasts with multi-oxide silicates that need to break more than one distinct metal–oxygen bond in their structure to complete their dissolution, and this affects the porous formation rates. In these cases, the dissolution rate is established by the release of the less reactive metal since the rupture of the different oxygen metal bonds can occur at speeds that differ in orders of magnitude.

The purpose of our work was to develop a new material obtained from the melt with a composition that includes calcium and magnesium phosphate, silicate bioactive phases, and a primary phase of magnesium oxide that forms pores when dissolved in SBF, creating the mechanical advantages of a dense material and the biological features of a porous scaffold. The microstructure upon different processing conditions and the bioactivity behavior through immersion in SBF were studied, and mechanical tests on the scaffolds were carried out.

## 2. Materials and Methods

### 2.1. Ceramic Preparation and Characterization

The following raw materials in the proportions, indicated in Table 1, were used: $SiO_2$ (purity: 99.8%, Alfa Aesar, Haverhill, Massachusetts, United States), calcium silicate (purity > 99%, Merck, Darmstaadt, Germany), MgO (purity > 99%, Merck, Darmstaadt, Germany), and $Ca_3(PO_4)_2$ (Carlo Erba, Barcelona, Spain).

**Table 1.** Composition (wt %) of the starting powder.

| MgO | CaSiO₃ | SiO₂ | TCP |
|------|--------|------|------|
| 36.6 | 18.7 | 14.04 | 30.7 |

Starting from a powdered material, we generated ceramic rods via cold isostatic pressing. After that, we carried out a sintering process in a Hobersal oven at 1300 °C for 12 h to obtain compacted solids. The crystalline nature of the bioceramic was identified by X-ray diffraction (XRD) (model D-Max/2500, RIGAKU) working at 40 kV and 80 mA, using $K_{\alpha1,2}$ radiation (1.5418 Å). The scanning was carried out between 5 and 80° (2 h) in 0.03° steps, counting for 1 s per step.

Microstructural characterization was performed in polished transverse and longitudinal cross sections of rods by means of back-scattered electron images obtained in a FE-SEM (Field Emission Scanning Electron Microscope model Carl Zeiss MERLIN, Jena, Germany). Quantitative analyses of Mg, P, Ca, Si, and O were conducted by means of the energy-dispersive X-ray spectroscopy (EDS) detector (INCA 350, Oxford Instruments, High Wycombe, UK) coupled with the FE-SEM. Specimens for this characterization were prepared using conventional metallographic procedures.

### 2.2. Laser Floating Zone Technique

Using the sintered rods as precursors, we grew crystalline bars by the laser floating zone (LFZ) technique using a $CO_2$ laser as a heating source (Blade 600, Electronic Engineering). This technique is based on focusing a laser beam on a precursor so that a small molten zone is established and moved along the sample to obtain a directionally solidified rod [20]. To eliminate the precursor porosity, a first densification step was applied at a pulling rate of 250 mm/h. The final directional solidification step was performed with the grown crystal traveling downwards to obtain bubble-free samples. Rods were grown at 50, 100, and 300 mm/h under a 50 rpm counter rotation of the solidified rod and the polycrystalline precursor. The solidified rods had a final diameter in the 2–2.5 mm range and a length of about 50 mm.

Phase formation and elemental composition of the phases at different pulling speeds were examined by FE-SEM.

### 2.3. Biodegradability Study

To study the biodegradability of the directionally solidified rods, they were cut into slices. After being washed in acetone, they were soaked in simulated body fluid (SBF), as proposed by Kokubo et al. [21,22], in polyethylene bottles kept at 37 °C under static conditions. The initial pH of the SBF was established between 7.2 and 7.4, in the range of normal pH of human plasma.

Disks were removed after 4 weeks and dried in air at room temperature. Sample surfaces and cross sections, before and after the exposure to the SBF, were examined by FE-SEM at 15 keV and EDS elemental microanalysis of calcium, magnesium, silicon, phosphorus, and oxygen were carried out.

The quantitative information of the ions released (Mg, Ca, P, and Si) into the SBF solution was determined by ICP using a Spectroblue TI FMT26 system.

### *2.4. Micro Hardness Test*

Micro indentations were performed on polished surfaces of unsoaked samples using a diamond Vickers indenter, in the form of a pyramid, on a microhardness tester Matsuzawa, MXT 70. The procedure followed for the preparation of samples for the Vickers hardness measurements was the same as that followed for FE-SEM observation. The samples were tested applying a load of 200 gf for 15 s. At least 10 valid indentations were made for each sample, and the data are presented as mean values with standard deviations.

## 3. Results and Discussion

### *3.1. Ceramic Analysis*

XRD and FE-SEM microstructural analysis of the ceramic precursors after sintering have been conducted, obtaining the results shown in Figures 1 and 2, respectively.

In Figure 1, there are peaks that correspond to periclase (MgO), forsterite ($Mg_2SiO_4$), and whitlockite ($Ca_{18}Mg_2H_2(PO_4)_{14}$). Because the powder holder is composed of $SiO_2$, peaks corresponding to this oxide were eliminated.

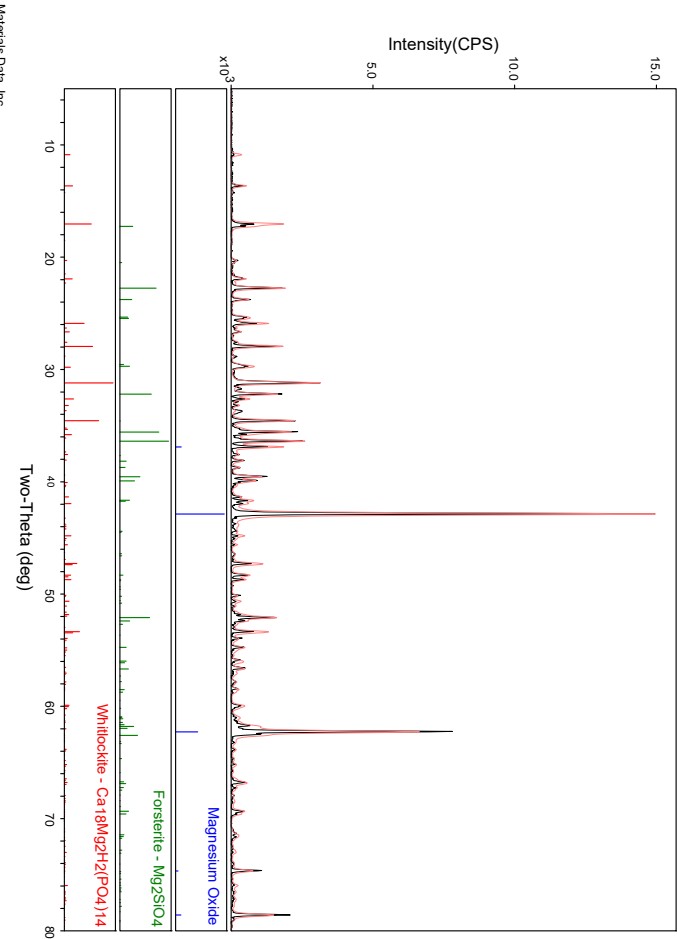

**Figure 1.** X-ray diffractogram of the ceramic material.

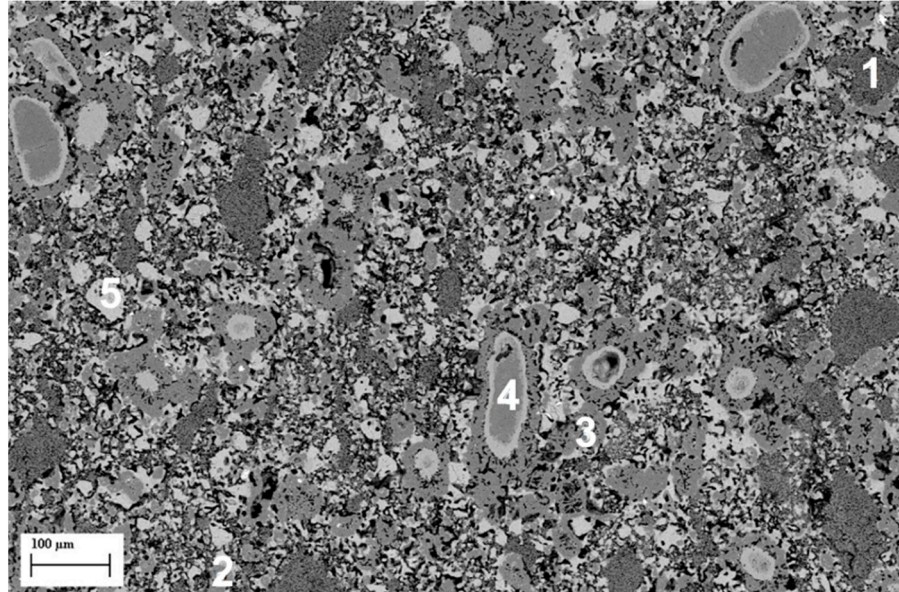

**Figure 2.** SEM (scanning electron microscope) image of the cross section of a sintered ceramic rod used as a precursor in the sample growth by the LFZ technique.

Figure 2 shows the microstructure of the sintered ceramic. It can be seen that, at this sintering temperature (1300 °C), reactions start to happen. For this reason, a light halo with a mixed oxide composition begins to form around $SiO_2$ grains (labeled as Phase 4, grey in the center of the picture). It is also possible to recognize the phases identified by XRD analysis: MgO (labeled as Phase 1, the darkest grey in the upper right corner of the picture), forsterite (labeled as Phase 3, dark grey), whitlockite (labeled as Phase 2, light grey), and tricalcium phosphate (TCP) (labeled as 5, the whitest phase). The composition of each phase is shown in Table 2. The chemical composition of a large region of the sample (General) corresponds in good approximation to the starting composition. In order to facilitate the identification of the different phases, the atomic percentages have been normalized to one of the ions.

**Table 2.** Elemental composition (atom %) of the main present phases in the sintered ceramic rods.

|  | O | Mg | Si | P | Ca | Phase |
|---|---|---|---|---|---|---|
| General | 59.17 | 21.63 | 7.02 | 5.21 | 6.27 | - |
| 1 | 1.0 | 1 | - | - | - | MgO |
| 2 | 62.9 | 2 | - | 16.9 | 20.3 | Whitlockite ($Ca_{18}Mg_2H_2(PO_4)_{14}$) |
| 3 | 3.8 | 1.8 | 1 | - | - | Forsterite ($Mg_2SiO_4$) |
| 4 | 2.0 | - | 1 | - | - | $SiO_2$ |
| 5 | 3.74 | - | - | 1 | 1.17 | Tricalcium Phosphate ($Ca_3(PO_4)_2$) |

*3.2. Microstructural Analysis of the Directionally Solidified Rods*

Starting from the ceramic bars, directionally solidified rods by the laser floating zone technique were grown. In order to analyze the influence of the solidification rate in the microstructure of the samples, different growth speeds were used. The samples named M50, M100, and M300 correspond to rods grown at 50, 100, and 300 mm/h, respectively. As the axial temperature gradient (G) at the solid–liquid interface were of the order of $5 \times 10^5$ K/m in the solid, the cooling rates (CR) were 7, 14, and 41.5 K/s for pulling rates (R) of 50, 100, and 300 mm/h, respectively (CR = G × R).

Figure 3 corresponds to a SEM picture of a transversal cross section of a sample grown at 50 mm/h. Three different phases can be observed: the black one corresponds to magnesium oxide (MgO) (1),

the dark gray corresponds to monticellite ($CaMgSiO_4$) (2), and the lighter gray is compatible with a solid solution of dicalcium magnesium silicate ($Ca_{2-x}Mg_xSiO_4$) (3) with phosphorus.

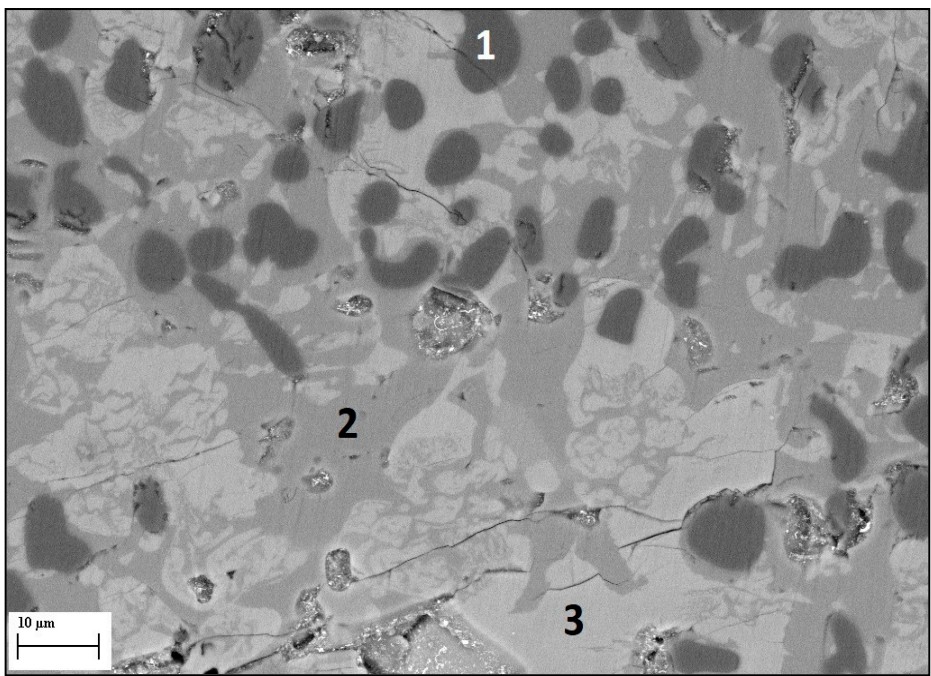

**Figure 3.** SEM image of a directionally solidified rod grown at 50 mm/h.

The identification of the phases and atomic percentages are shown in Table 3. In Phase 3, the Mg ion replaced the position of the Ca ion, and this formed a solid solution. As more than 15% of Ca was substituted by Mg, it is expected that the crystallinity of this phase cannot be retained, which forms a vitreous network where phosphorus can be incorporated.

**Table 3.** EDS analysis (atom %) for directionally solidified rods at 50 mm/h.

| Spectrum | O | Mg | Si | P | Ca | Phase |
|----------|-------|-------|-------|------|-------|-------|
| General | 57.89 | 16.29 | 10.52 | 1.95 | 13.35 | |
| 1 | 50.67 | 49.33 | - | - | - | MgO |
| 2 | 57.71 | 13.96 | 14.84 | 0.52 | 12.98 | Monticellite ($CaMgSiO_4$) |
| 3 | 59.04 | 3.98 | 12.28 | 3.12 | 21.57 | $Ca_{2-x}Mg_xSiO_4$ + P |

Figure 4 shows a SEM cross-section view of the sample grown at 100 mm/h. In this micrograph, four phases can be observed: the dark one corresponds to magnesium oxide (MgO) primary phase (1), and the others correspond to monticellite ($CaMgSiO_4$), akermanite (3) ($Ca_2MgSi_2O_7$), and TCP (4). Part of the TCP formed a eutectic constituent with monticellite (2), and the akermanite phase contained phosphorus that was dissolved in its structure. The atomic percentages of the phases are shown in Table 4.

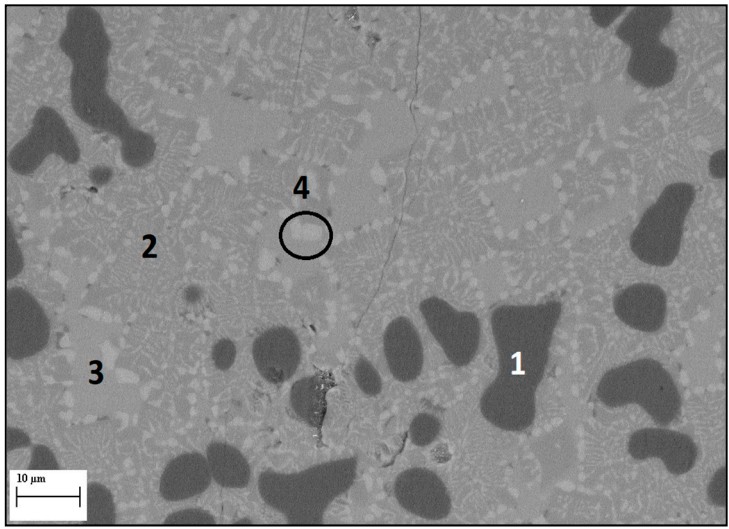

**Figure 4.** SEM image of a directionally solidified rod grown at 100 mm/h.

**Table 4.** EDS analysis (atom %) for directionally solidified rods at 100 mm/h.

|  | **O** | **Mg** | **Si** | **P** | **Ca** | **Phase** |
|---|---|---|---|---|---|---|
| General | 58.20 | 17.54 | 9.97 | 9.97 | 11.30 |  |
| 1 | 50.94 | 48.89 | - | - | - | MgO |
| 2 | 57.93 | 13.64 | 11.4 | 3.32 | 13.71 | Monticellite + TCP (eutectic constituent) |
| 3 | 60.78 | 6.54 | 15.35 | 3.47 | 13.86 | Akermanite + P (solid solution) |
| 4 | 60.16 | - | - | 10.15 | 19.13 | TCP |

Monticellite has been considered a candidate for bone replacement [23,24] because of its good cytocompatibility, osteogenic activity, and antibacterial and anti-biofilm properties. Akermanite has shown bioactive properties in both in vivo and in vitro conditions [25,26].

Figure 5 shows a SEM cross-section view of the sample grown at 300 mm/h. Three main phases, whose compositions are indicated in Table 5, can be identified.

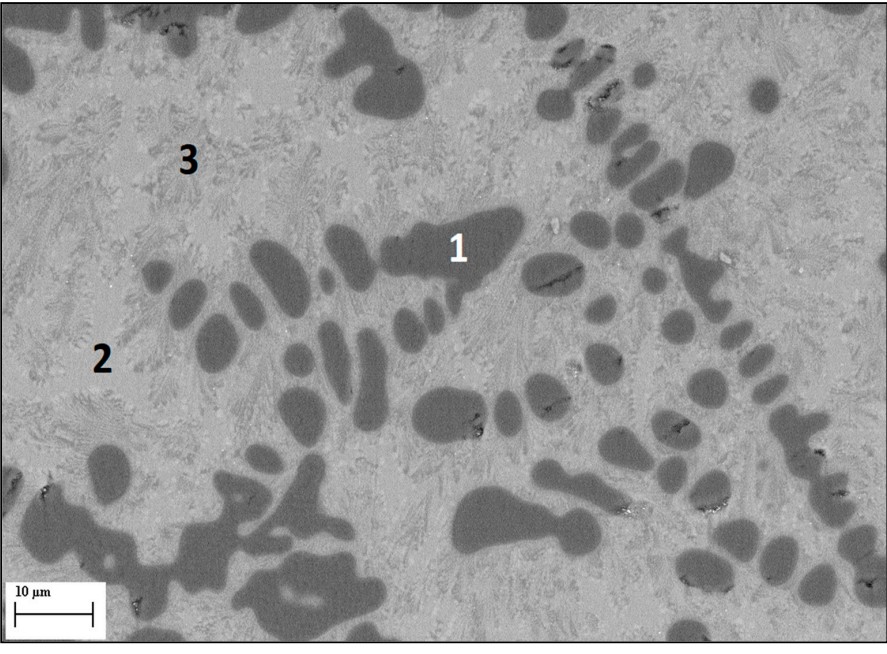

**Figure 5.** SEM micrograph of a directionally solidified rod grown at 300 mm/h.

**Table 5.** EDS analysis (atom %) for directionally solidified rods at 300 mm/h.

|  | O | Mg | Si | P | Ca | Phase |
|---|---|---|---|---|---|---|
| General | 57.66 | 21.81 | 8.27 | 3.22 | 9.05 | |
| 1 | 50.52 | 49.33 | - | - | - | MgO |
| 2 | 60.78 | 6.31 | 15.61 | 3.18 | 14.36 | Akermanite + P (solid solution) |
| 3 | 59.47 | 13.06 | 10.75 | 4.79 | 11.93 | Monticellite + TCP (eutectic constituent) |

It is possible to detect small phases surrounding the akermanite, and these phases can be assigned to TCP based on a comparison with the sample grown at 100 mm/h. In fact, both microstructures are similar except for the smaller size of the phases in the case of the sample grown at a faster rate, as expected.

At a faster growing rate (300 mm/h), the general composition is more similar to the starting one than that of samples grown at slower rates (100 and 50 mm/h). This fact might be due to the loss by evaporation of the most volatile elements, mainly P, and this loss is higher at lower growth speeds. The loss of phosphorus by evaporation could explain the impossibility of TCP formation in samples grown at lower speeds.

### 3.3. Micro Hardness Analysis

Hardness values were obtained from Vickers micro hardness tests on polished cross sections of the unsoaked samples. These values are depicted in Table 6 and compared with the hardness of different bioceramics and bone tissues.

**Table 6.** Vickers hardness comparison of different materials used as bone substitutes. M50, M100, and M300 correspond to samples grown at 50, 100, and 300 mm/h.

| Sample | Growing Speed (mm/h) | Vickers Hardness (GPa) |
|---|---|---|
| Bone [2] | - | 0.42 |
| Enamel [27] | - | 3.3–3.6 |
| TCP-5wt % MgO composites [28] | - | 4.5 |
| $CaMg(SiO_3)_2–Ca_3(PO_4)_2$ [3] | - | 4.1 ± 0.4 |
| $CaSiO_3–Ca_3(PO_4)_2$ [3] | - | 5.1 ± 0.4 |
| $CaSiO_3–CaMg(SiO_3)_2–Ca_3(PO_4)_2$ (eutectic glass) * | - | 4.13 ± 0.35 |
| M50 | 50 | 4.63 ± 0.36 |
| M100 | 100 | 5.30 ± 0.43 |
| M300 | 300 | 5.02 ± 0.33 |

* Grown by the LFZ technique for this work.

Poor mechanical properties limit the applications of ceramics with good bioactivity. In this sense, the development of Mg–Ca-based ceramics with increasing Mg content (8.3% for akermanite and 14.3 for monticellite) has led to the improvement of the mechanical properties of materials for bone defect repair. As the bond energy of the Mg–O is higher than that of the Ca-O due to their differences in ion radius, the presence of magnesium silicates may be beneficial for the mechanical strength of the scaffolds. In the table, we can observe that the hardness obtained for the present samples are close to the values of other materials used for the same purpose. For this reason, we can conclude that the hardness of this new material is in the required range of a bone substitute.

### 3.4. Biodegradable Study of the Solidified Rods

To study the bioactive behavior of the directionally solidified rods, we placed some discs of different samples in SBF. After being soaked for 4 weeks, we analyzed them via SEM. Figure 6a shows the effect of SBF in a directionally solidified rod grown at 50 mm/h (M50). An extended area of the surface in contact with the fluid is presented in Figure 6b. In those images, we can observe the dissolution of one of the phases, which corresponds to the MgO phase.

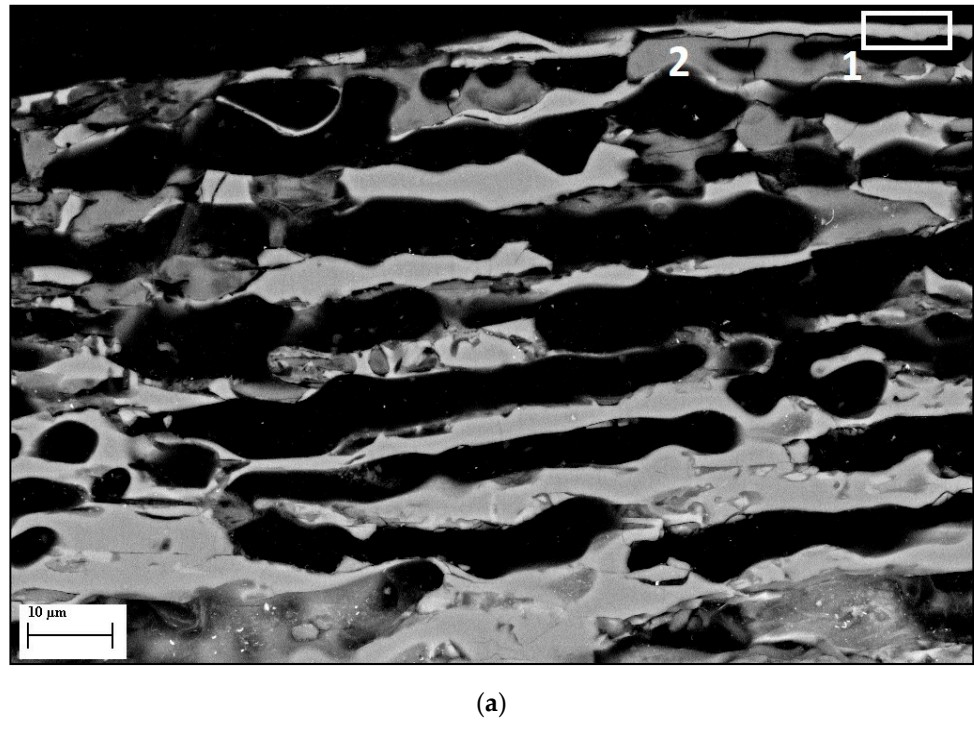

(**a**)

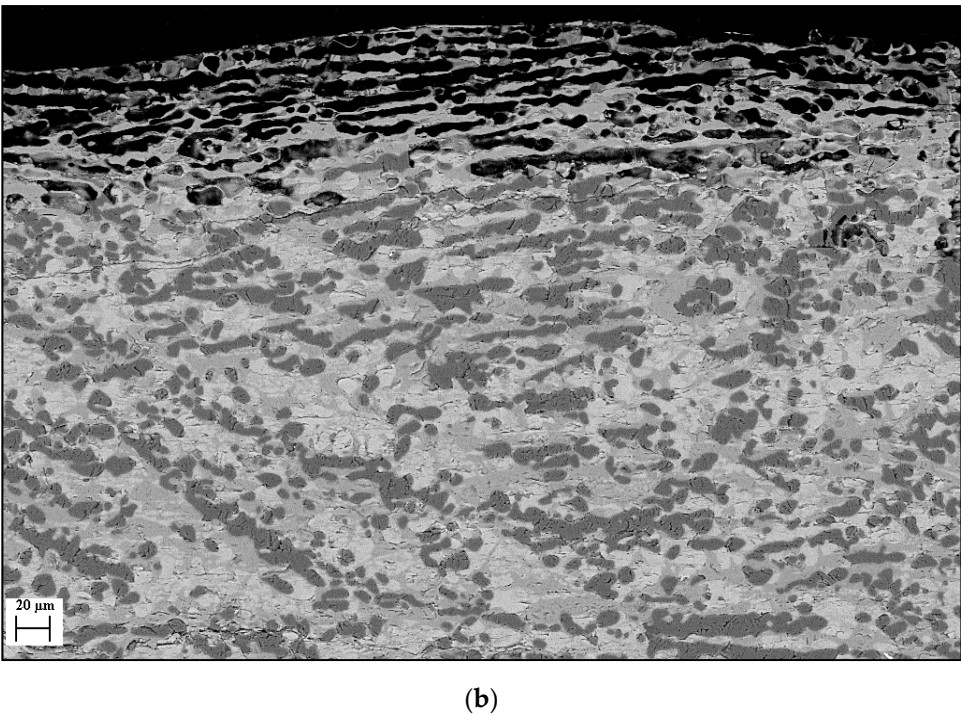

(**b**)

**Figure 6.** SEM micrographs of a directionally solidified rod grown at 50 mm/h after soaking in the simulated body fluid (SBF) solution for 28 days: longitudinal cross section (**a**) and an extended area of the sample surface in contact with the fluid (**b**).

As adjacent MgO phases dissolve, an open porosity forms, and this provides transport channels for the migration of protons and $Mg^{2+}$ ions. In this case, the contiguity and volume fraction of the MgO phases are important determining factors in the penetration depth of the dissolution reaction.

No external HA layer that would block the dissolution of the MgO was formed. For this reason, the dissolution of the MgO phase could continue, which gave rise to a porous scaffold with

channels between 5 and 10 μm in thickness. MgO dissolution is controlled by chemical reactions (Equations (1)–(3)) that involve the dissolution of MgO in a liquid medium to produce $Mg^{2+}$ and $OH^-$. As a consequence of water molecules, an intermediate brucite product was generated and subsequently dissociated into $Mg^{2+}$ and hydroxyl ions that form water by protonation.

$$MgO\ (s) + H_2O\ (l) \rightarrow MgOH^+\ (aq) + OH^-\ (aq) \tag{1}$$

$$MgOH^+\ (aq) + 2H_2O\ (l) \rightarrow Mg(OH)_2\ (l) + H_3O^+\ (aq) \tag{2}$$

$$Mg(OH)_2\ (l) \rightarrow Mg(aq)^{2+} + 2OH^-\ (aq) \tag{3}$$

In the zone where MgO dissolution occurred, the monticellite phase (1) remained unchanged, while the composition of the other phase (2) changed, losing silicon and incorporating phosphorus to a composition similar to that of the monetite ($CaHPO_4$), as indicated in Table 7.

**Table 7.** EDS analysis (atom %) for directionally solidified rods grown at 50 mm/h after soaking in the SBF solution for 28 days.

|   | O | Mg | Si | P | Ca | Phase |
|---|---|---|---|---|---|---|
| 1 | 61.51 | 13.20 | 13.22 | 0.99 | 11.08 | Monticellite |
| 2 | 70.01 | 3.09 | 0.30 | 14.05 | 12.56 | Monetite |

These compositional changes of the phases occurred by interaction of the sample surface with the liquid medium and have an influence on the final composition of the fluid. In Table 8, we present the ICP results of the SBF analysis after being in contact with a directionally solidified rod compared with the concentration values in the SBF before the bioactivity test.

**Table 8.** Ion concentration of SBF (simulated body fluid) before and after 28 days of sample inmersion. Results in mM.

|   | Ca | Mg | P | Si |
|---|---|---|---|---|
| Before Bioactivity | 2.5 | 1.5 | 1 | - |
| After Bioactivity | 2.2 | 3.5 | 0.91 | 0.22 |

As can be seen in Table 8, during the reaction of the ceramic composite in SBF, the calcium and phosphorus ion concentrations did not increase during exposure, although an increase in the magnesium and silicon ion concentrations can be observed. This fact indicates the surface dissolution of the MgO primary phase and release to the fluid of $Si^{4+}$ from the soluble amorphous silicate phase (Table 3, Phase 3).

Both the increase in the concentration of Mg ion in SBF after the bioactivity test and the porous structure formation evidence the dissolution of the MgO primary phase. This mechanism led to the "in situ" formation of a porous structure that could reproduce the bone structure and its mechanical properties.

The increase in Mg and Si ion concentration is attributed to the exchange of Mg and Si ions from the sample with $H^+$ ions from the fluid. The MgO dissolution could proceed by the breaking of the relatively weak ionic divalent metal–oxygen bonds, which liberated the Mg ions directly into solution. This fast dissolution of magnesia contrasts to the behavior of other multi-oxide phases that require the breaking of more than one type of metal–oxygen bond. In these cases, the dissolution mechanism involves the sequential breaking of the bonds, which follows an order according to their reactivity. Some studies of the forsterite-rich olivine dissolution mechanisms have been reported by Oelkers et al. [19]. They concluded that olivine dissolution rates are strongly influenced by pH, water activity, and mineral–fluid interfacial surface area.

There is no evidence that a silica-rich layer was formed. When this layer, due to the reaction of Si with OH⁻, was formed on the sample, the surface acted as a nucleation agent for HAp formation. In this case, the increase in Mg in the fluid medium can reduce the rate of a stable apatite phase formation, as reported by Vallet-Regi [29]. Moreover, the decrease in P ion concentration in the fluid was consistent with the transformation of the $Ca_{2-x}Mg_xSiO_4{}^+P$ phase to a P-rich and Si-depleted phase, while the monticellite phase remained unaltered.

The longitudinal cross section and surface morphology of the sample grown at 100 mm/h after soaking in SBF for 28 days are shown in Figure 7a. The presence of a large number of pores at different sizes significantly increased the surface roughness and total porosity, as shown in Figure 7b. This morphology contributed beneficially to the process of scaffold integration and influenced the bone healing rate. The composition of the phases in the interaction zone with SBF is given in Table 9. In the active region pores, monticellite and a new phase with a composition compatible with monetite can be identified.

**Table 9.** EDS analysis (atom %) for directionally solidified rods grown at 100 mm/h after soaking in the SBF solution for 28 days.

|   | O | Mg | Si | P | Ca | Phase |
|---|-----|-----|-----|-----|-----|-------|
| 1 | 70.2 | 4.14 | 0.35 | 13.45 | 11.94 | Monetite |
| 2 | 58.10 | 14.10 | 14.38 | 0.81 | 12.61 | Monticellite |

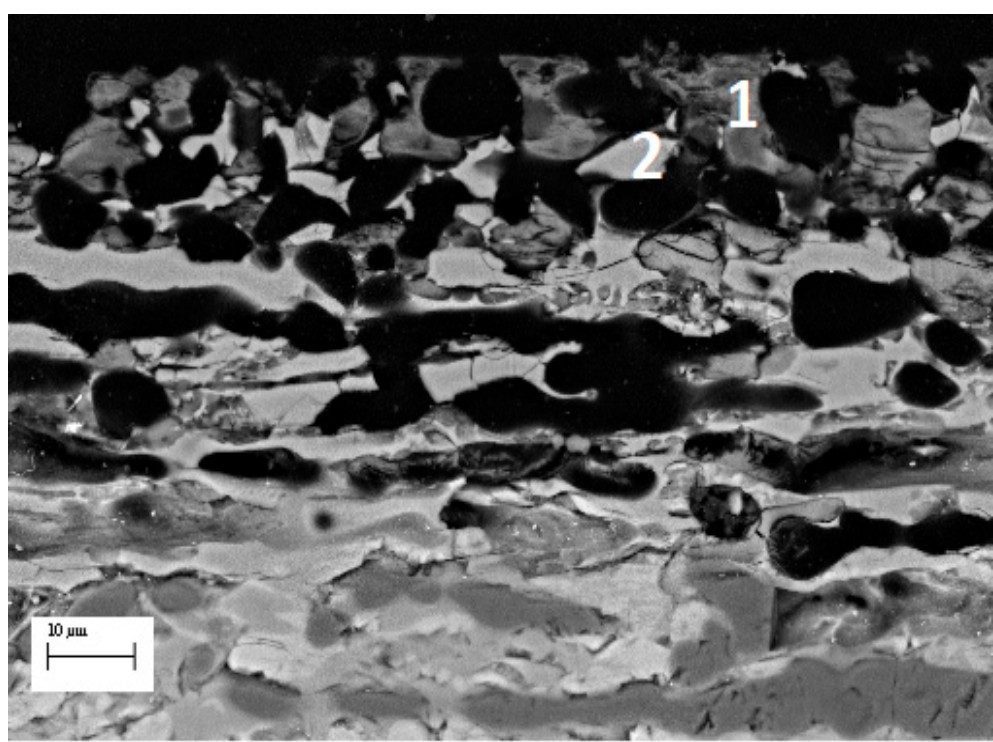

(**a**)

**Figure 7.** *Cont.*

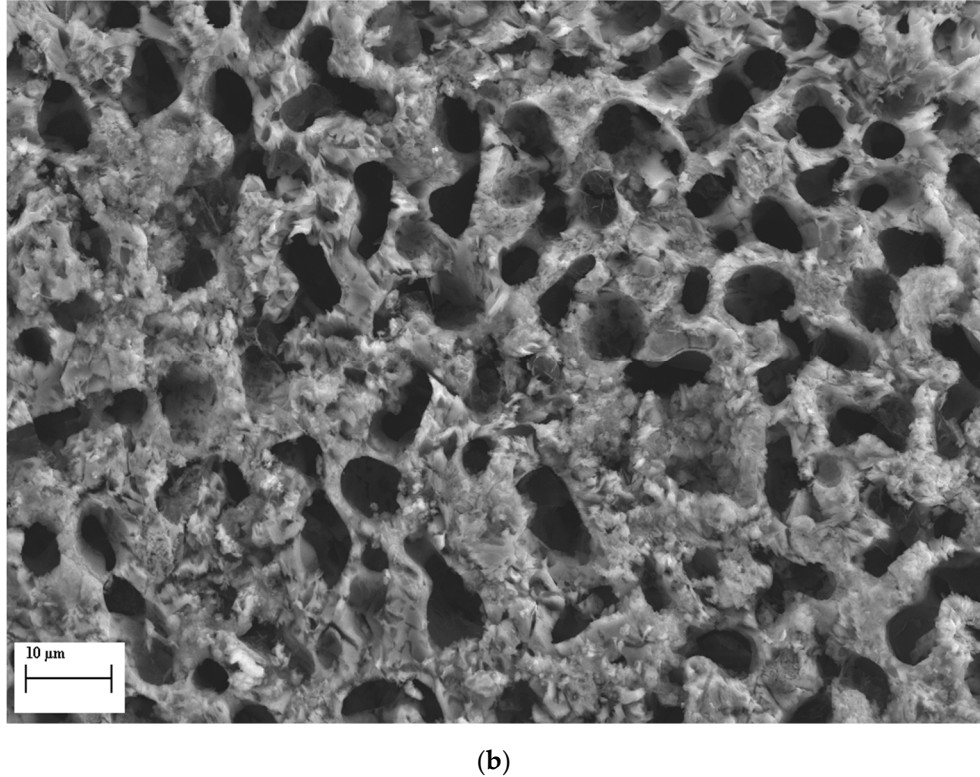

(**b**)

**Figure 7.** SEM image of a sample grown at 100 mm/h after soaking in the SBF solution for 28 days: longitudinal cross section (**a**) and disc surface (**b**).

A similar behavior was observed in samples grown at 300 mm/h. The FE-SEM micrograph of the polished longitudinal cross section of the sample after soaking in SBF for 28 days is shown in Figure 8a. The surface of the sample was eroded by the dissolution of the MgO phases in the SBF during the period of immersion forming a porous structure layer. The depth of the porous layer at this time was about 20–30 μm.

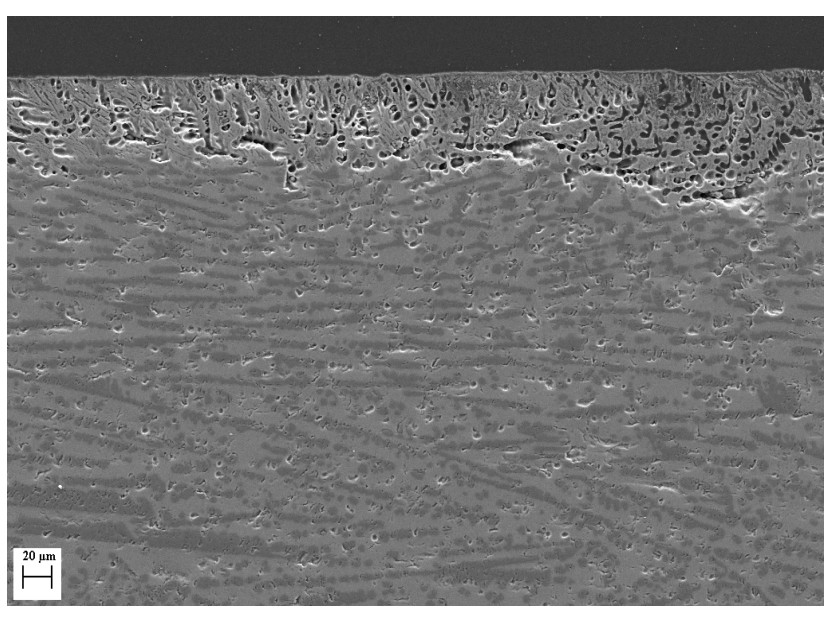

(**a**)

**Figure 8.** *Cont.*

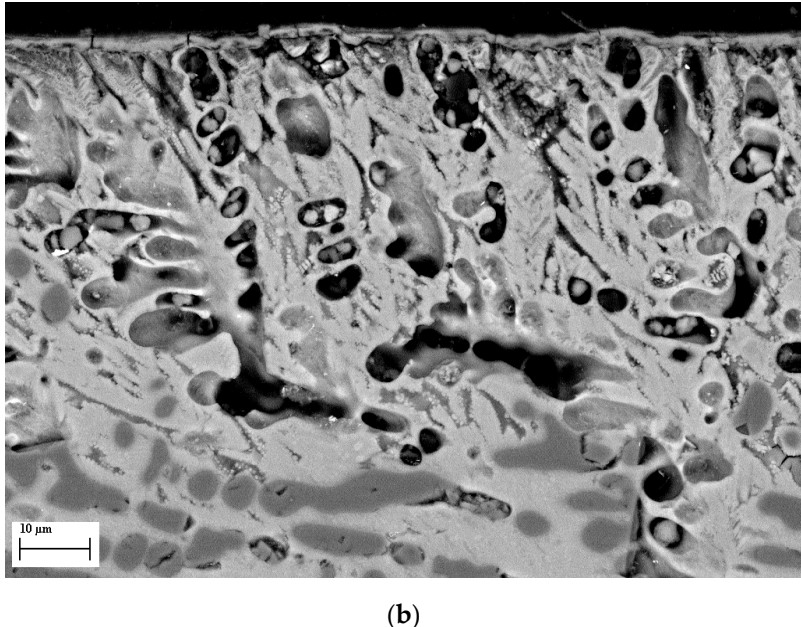

(**b**)

**Figure 8.** SEM image of a sample grown at 300 mm/h after soaking in the SBF solution for 28 days: longitudinal cross section (**a**) and a detailed zone of the sample surface in contact with the fluid (**b**).

The formation of a thin submicrometric layer was observed on the surface of the sample as shown in Figure 8b. This layer was determined to be bone-like apatite, although from EDS microanalysis the Ca/P ratio was 1.26, lower than that of the hydroxyapatite.

## 4. Conclusions

At the end of this work, we could establish the following conclusions:

- Crystalline rods with compositions in the $CaO–SiO_2–MgO–P_2O_5$ system were grown via the LFZ technique at speeds of 50, 100, and 300 mm/h, and an MgO primary phase was obtained with other bioactive phases at all speed conditions.
- The surface of the crystalline rods (M50, M100, and M300) presented a response in contact with the SBF, and this response consisted of the dissolution of magnesium oxide phases and the transformation of some of the other phases. The dissolution of MgO phases generated a porous layer with interconnected pores that acted as transport channels for the liquid medium to continue MgO leaching inside the matrix.
- Samples grown at 50 and 100 mm/h after soaking in SBF for four weeks did not form HAp layer, and this favored the dissolution of the MgO phases. The release of biocompatible $Mg^{2+}$ ions acted as an inhibitor of hydroxyapatite crystal growth suppressing unwanted crystallization in vivo.
- The micro hardness of the unsoaked samples lies between 4.63 and 5.30 GPa, which is comparable to other bioceramics used in bone repair. Porosity formation by the leaching of magnesium oxide in SBF can reduce the hardness, bringing it closer to that of natural bone.
- As the material is able to generate a porous scaffold, in order to conclude if this new Mg-based formulation can be used as a bone replacement, cell tests and in vivo experiments are needed.

**Author Contributions:** J.I.P. and L.G. conceived and planned the experiments. J.G., L.G., M.D.-P. and D.S. carried out the experiments. J.I.P. contributed to sample preparation and characterization. All authors contributed to the interpretation of the results and provided critical feedback. J.I.P. wrote the paper with input from all authors. All authors have read and agreed to the published version of the manuscript.

**Funding:** This project has been partially funded by the PIT2 program of the University of Murcia's own research plan. Fundación Séneca grants No 20647/JLI/18 and the European Union's Horizon 2020 research and innovation program under the Marie Skłodowska-Curie IF No 795630 are also acknowledged.

**Acknowledgments:** The authors would like to acknowledge the use of Servicio General de Apoyo a la Investigación-SAI, Universidad de Zaragoza.

**Conflicts of Interest:** The authors declare no conflict of interest.

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
