# Peer review of "Generation of a Porous Scaffold with a Starting Composition in the CaO–SiO2–MgO–P2O5 System in a Simulated Physiological Environment"

_applsci, doi:10.3390/app10010312_

Round 1

Reviewer 1 Report

The manuscript reports on a new material obtained from the melt with a composition that includes calcium and magnesium phosphate, silicate bioactive phases and a primary phase of magnesium oxide. The latter forms pores when dissolved in simulated body fluid and could be the basis for the biological features of a porous scaffold. Evaluated were microstructures formed by diverse processing conditions, bioactivity following their immersion in simulated body fluid and mechanical tests of the scaffolds.

Overall remarks

This type of glass was described before by others. You now used the laser floating Zone technique to produce diverse rods. Here you address the leaching of Mg and the formation of pores. Since these pores are very small and in the micrometer range, you deal with microporosity. Unfortunately, any in vitro or in vivo tests are missing at all.

Specific remarks

In the introduction, the meaning of microporosity for bone regeneration should be introduced.

In figure 2, it is not easy to find the corresponding numbers.

Overall, it is a very descriptive work.

Author Response

The authors thank reviewer 1 for his accurate insights on our manuscript.

We agree that the study must proceed by performing in vitro cell tests. We will consider this suggestion for future work.

Comments from reviewer #1

In the introduction, the meaning of microporosity for bone regeneration should be introduced.

Thank you. This is an important point not sufficiently addressed in the introduction.

Navalon et al. (C. Navalón, P. Mazón, P.N. De Aza, Eutectoid dicalcium silicate-Nurse's A ceramic scaffold: Processing and in vitro bioactivity, Ceramics International 45 (2019) 21716-21724) reported that pores > 300 μm improve cell migration, osteoblast proliferation/spreading and new vascular tissue formation through the structure. However, the presence of microporosity may enhance ionic exchange at the scaffold surface that comes into contact with SBF, which improves scaffold bioactivity.

It has been also reported that surface micropore modulates the adhesion, proliferation, and pheno- typic expression of MG-63 osteoblast-like cells (R. Detsch, O. Guillon, L. Wondraczek, A.R. Boccaccini, Initial attatchment of rMSC and MG-63 cells on patterned bioglass® substrates, Adv. Eng. Mater. 14 (2012) B38–B44). Then, a synergistic combination between macro and microporosity is desirable to have good osteocondutive and osteoinductive properties.

In the new version of the manuscript we have included a comment regarding the effect of microporosity on bone regeneration, taken into account that microporosity imparts biomimetic features to the biomaterial and is essential for the interactions between the cells and the matrix contributing to effects like a larger surface area and enhancing cell adhesion to the substrate.

In figure 2, it is not easy to find the corresponding numbers.

To facilitate the identification of the phases in Figure 2, the description of these phases in the text and the picture have been improved.

Reviewer 2 Report

In my opinion, the article is well prepared and after some corrections can be printed.

However, the Figure 1.  can be improved as well as the descriptions of the drawings on white backgrounds should be corrected (no description in Figure 8b).

Author Response

Answer to referee 2.

We deeply appreciate the valuable suggestions of reviewer #2.

Comments from reviewer #2:

First, I would like to thank the reviewer for his effort and careful reading of the manuscript as well as his wise suggestions.

1. Figure 1 has been improved. I hope that the new figure allows a clearer description of the phases present in the starting ceramic.

2. The reviewer is absolutely right pointing out that the description of Figures 6, 7 and 8 is not good. We have included some piece of information missed relative to figure 8 b and specified the description of the others. These changes are highlighted in the text.

I hope this improved version of the manuscript meets the quality standards of the journal.

Reviewer 3 Report

This study is focus on a new composite ceramic material was obtained by Laser Floating Zone (LFZ) technique.  I think it's an interesting idea. However, I have some question about method and result.

In section 2.2, directionally solidified rod was prepared through densification and directional solidification. The working parameter name must unite. If author want to choose one from "pulling rate" or "solidification speeds". I think the pulling rate is more suitable.  On the other hand, the solidified rods had a final diameter in the range 2–2.5 mm.  What is the length of the rod? 

In section 2.3, how much is replacement times of solution in SBF immersion peroid? If not replaced, the pH value of the solution will become alkaline and accelerate the dissolution of magnesium oxide. Therefore, the porous structure formation on rod surface may be slower than described on this study.

The title, “The bioactivity study of the solidified rods” may be modified to “the biodegradable study of the solidified rods”. Because the result was show biodegradable more than bioactivity on section 3.4.

Effect of pulling rate will affect solidification speeds of ceramic rod under atmosphere environment. Therefore, phase composition of ceramic rod will be changed. Is it possible to compare phase changes with cooling rate of rod?

Author Response

Answer to referee 3.

We deeply appreciate the valuable suggestions of reviewer #3.

Comments from reviewer #3:

1.In section 2.2, directionally solidified rod was prepared through densification and directional solidification. The working parameter name must unite. If authors want to choose one from "pulling rate" or "solidification speeds". I think the pulling rate is more suitable.  On the other hand, the solidified rods had a final diameter in the range 2–2.5 mm.  What is the length of the rod?

Thank you. We have unified the working parameter in the text using “pulling rate” that corresponds to the parameter controlled by the LFZ technique, as the reviewer precisely indicates. The length of the rods is typically 50 mm. We have included this piece of information in the text.

2. In section 2.3, how much is replacement times of solution in SBF immersion period? If not replaced, the pH value of the solution will become alkaline and accelerate the dissolution of magnesium oxide. Therefore, the porous structure formation on rod surface may be slower than described on this study.

 Thank you. This a very interesting point that deserves to be clarified. The solution was not stirred and not replaced during soaking time so that, a change to more alkaline pH is expected while the dissolution of the magnesium oxide proceeds. Some authors have studied the effect of acid concentration in the dissolution kinetic of MgO dissolution concluding that the rate of chemical dissolution of MgO accelerated with increase in H+concentration ions (A. Fedorockova and P. Raschman, Effects of pH and acid anions on the dissolution kinetics of MgO, Chemical Engineering Journal, 143 (2008) 265-272). Therefore, the rate of transfer of magnesium to the solution is significantly affected by pH in the way that a pH increase due to MgO dissolution results in lower dissolution rate. As a conclusion it is expected that under stirring or dynamic flow the dissolution rate would be higher than in static conditions.

3. The title, “The bioactivity study of the solidified rods” may be modified to “the biodegradable study of the solidified rods”. Because the result was show biodegradable more than bioactivity on section 3.4.

 We totally agree with this comment. This change is included in the new version.

4. Effect of pulling rate will affect solidification speeds of ceramic rod under atmosphere environment. Therefore, phase composition of ceramic rod will be changed. Is it possible to compare phase changes with cooling rate of rod?

Yes. Pulling rate (R) affects solidification speed. This is a growth parameter that we can control during rod growth but it may of interest only to others authors using the same growth technique. As reviewer very properly indicates the cooling rate (CR) is a more useful growth parameter. As the axial temperature gradient (G) at the solid-liquid interface were of the order 5 x 105K/m in the solid, the cooling rates are 7 K/s, 14 K/s and 41,5 K/s for pulling rates of 50 mm/h, 100 mm/h and 300 mm/h, respectively (CR = G x R). We have included the cooling rate values in the text.